# Worker-Born Males Are Smaller but Have Similar Reproduction Ability to Queen-Born Males in Bumblebees

**DOI:** 10.3390/insects12111008

**Published:** 2021-11-09

**Authors:** Huiyue Zhao, Yanjie Liu, Hong Zhang, Tom D. Breeze, Jiandong An

**Affiliations:** 1Key Laboratory for Insect-Pollinator Biology, Ministry of Agriculture and Rural Affairs, Institute of Apicultural Research, Chinese Academy of Agricultural Sciences, Beijing 100093, China; zhaohuiyue1124@163.com (H.Z.); liuyanjie@caas.cn (Y.L.); zhanghong@caas.cn (H.Z.); 2Centre for Agri-Environmental Research, School of Agriculture, Policy and Development, Reading University, Reading RG6 6AH, UK; t.d.breeze@reading.ac.uk

**Keywords:** *Bombus terrestris*, queenless micro-colony, males, copulation, sperm, colony foundation

## Abstract

**Simple Summary:**

In bumblebee colonies, the authority of male reproduction is not entirely controlled by the queen. Workers who lose mating ability could also produce haploidy males. The conflict over the reproduction of males in bumblebee colonies becomes more evident compared to the higher eusocial honeybees. However, the reproductive characteristics of worker-born males in bumblebees remain unclear. Here, we evaluated the body size and reproductive ability of males produced by worker and queen in *Bombus terrestris*. Smaller size and slighter individuals’ weight but equally high sperm viability was found in worker-born males compared to that of queen-born males. Moreover, worker-born males performed as excellently as queen-born males in copulation and colony development of the mated queens, such as queen egg laying, colony foundation, and colony size. This study contributes the new knowledge of the reproductive biology of eusocial bees.

**Abstract:**

Queen-worker conflict over the reproduction of males exists in the majority of haplodiplioidy hymenpteran species such as bees, wasps, and ants, whose workers lose mating ability but can produce haploid males in colony. Bumblebee is one of the representatives of primitively eusocial insects with plastic division labor and belongs to monandrous and facultative low polyandry species that have reproductive totipotent workers, which are capable of competing with mother queen to produce haploid males in the queenright colony compared to higher eusocial species, e.g., honeybees. So, bumblebees should be a better material to study worker reproduction, but the reproductive characteristics of worker-born males (WMs) remain unclear. Here, we choose the best-studied bumblebee *Bombus terrestris* to evaluate the morphological characteristics and reproductive ability of WMs from the queenless micro-colonies. The sexually matured WMs showed smaller in forewing length and weight, relatively less sperm counts but equally high sperm viability in comparison with the queen-born males (QMs) of the queenright colony. Despite with smaller size, the WMs are able to successfully mate with the virgin queens in competition with the QMs under laboratory conditions, which is quite different from the honeybees reported. In addition, there was no difference in the colony development, including the traits such as egg-laying rate, colony establishment rate, and populations of offspring, between the WM- and the QM-mated queens. Our study highlights the equivalent reproductive ability of worker-born males compared to that of queens, which might exhibit a positive application or special use of bumblebee rearing, especially for species whose males are not enough for copulation. Further, our finding contributes new evidence to the kin selection theory and suggests worker reproduction might relate to the evolution of sociality in bees.

## 1. Introduction

Bumblebee species possess a variety of traits such as tongue length, buzz-pollinating behavior, and adaptation to low pollen or low environmental temperature foraging, making them efficient pollinators in agricultural and natural ecosystems [1,2,3,4,5,6,7,8]. Like most Hymenoptera, Bumblebees have a haplodiploidy sex-determination system, a key feature in the evolution of sex determination in Hymenoptera [9,10]. Many bumblebee species are primitively eusocial, with a unique queen who produces diploid females arising from fertilized eggs and haploid males arising from unfertilized eggs [11]. Bumblebee workers also take on distinct tasks within an annual colony, but the division of labor is more plastic compared to the higher eusocial honeybees [12]. Unmated bumblebee workers can still produce haploid male offspring [9]. Therefore, kin selection theory predicts that the daughter workers will compete with the mother queens for male production [13,14,15]. In *Bombus terrestris,* one of the most common bumblebee species, studies into the conflict between queen and workers indicate that queens maintain the dominance of male production [16,17,18,19]. This reproductive competition is closely related to several interactions between queens and workers (pheromonal signaling, methylation, Dufour’s gland secretion, nest wax, and attacking behavior) [20,21,22,23,24,25,26,27,28]. Similar queen-worker conflict over the production of males exists in the majority of social Hymenoptera species, with similar queen dominance in male production [29,30,31,32]. However, reproductive bumblebee workers show more similar behavior and physiology to queens than similar nonreproductive workers in eusocial species [33]. Ultimately, approximately 5% of males during the competition phase are produced by reproductive workers in queenright monandrous *B. terrestris* colonies [18,19].

Males emerge at the switch point during the reproductive season of the life cycle in the annual bumblebee colony [34]. *B. terrestris* males sexually mature 6−22 days after emergence, and 9-day-old males have the highest mating success [35,36]. The reproductive ability of males, such as their mating ability and sperm quality, makes a fundamental contribution to the copulation of queens and the establishment of offspring colonies [37,38]. In other social hymenoptera, the quality of males’ sperm is known to influence the offspring colony. Several parameters, such as sperm viability and sperm plasma membrane integrity, have been assessed by multiple testing approaches in the honeybees *Apis m**ellifera* [39,40]. The successful laying of fertilized eggs by honeybee queens in the process of the colony establishment requires the proportion of live sperm to reach the level of more than 50% [41]. Sperm quality can also influence traits of *B. terrestris* queens, such as hibernation success, queen longevity, and fitness [42]. Sperm quality is influenced by environmental factors such as nutrition, temperature, and the inherent characteristics of males, such as body size [43]. The body size of males, in particular, affects mating success and mating duration; a short duration is associated with the delay of the establishment of colonies [44].

In honeybees, the different origins (worker- or queen-born) of male-destined eggs lead to somatotype dimorphism of males [45]. Worker-born male (WMs) honeybees have lighter weight and significantly smaller, in forewing length, hind leg length, and head width, than and significantly inferior sperm counts, sperm concentration, the weight of mucus glands, and seminal vesicles compared to queen-born males (QMs) [46,47,48]. In addition, small males possess relatively poor visual and flying power that might limit the access of males to honeybee queens [49,50], resulting in as little as half the number of copulations that would be expected given the number of flights [51]. Beyond mating success, small males that have a mating opportunity still have lower paternity shares than expected when competing with the QMs in vivo [52].

Bumblebee workers can produce a high percentage of WMs even in queenright colonies during its competition phase. However, the sperm characteristics and the contribution of the WMs to the offspring colony are still unclear. Here, we compared the weight and forewing length, sperm counts, sperm viability and mating success of WMs and QMs, and populations of offspring colonies that are established by the queens copulating with WMs and QMs aimed to explore the feature and the reproductive ability of WMs in bumblebees.

## 2. Materials and Methods

### 2.1. Rearing of Male and Queen Bumblebees

Fifteen bumblebee *B. terrestris* colonies were bought from Biobest Group NV (Shouguang, China) in October 2020 and were reared in the rearing room (28 ± 1 °C, 50% ± 55% RH, corresponding to the principles of the commercial rearing [53] with little modification) at Institute of Apicultural Research, Chinese Academy of Agricultural Sciences, Beijing, China. Each colony had one vigorous queen and 40−50 workers and were in the stage that before the switch point when males were not yet produced. In all the colonies, males and queens were fed pollen paste and sugar syrup (50% sugar content, *w*/*w*) following the methodology described in Zhang [54].

Because identifying male brood is complicated by the irregular shape of bumblebee colonies, with scattered and characterless cells, we obtained WMs through capturing workers from queenright bumblebee colonies to establish queenless micro-colonies [55]. Ten bumblebee colonies were chosen to establish these queenless micro-colonies to produce the WMs [55]. From each colony, four micro-colonies were started, each established from three workers of similar age from the original colony (Figure A1a). Thus, a total of 40 micro-colonies were used in this study to obtain the WMs. Then, the original ten colonies were continually reared as queenright colonies to produce QMs. The laying of QM eggs starts about 23 days after the establishment of the colony, which has been regarded as the switch point. The reproductive competition between mother queen and daughter workers starts from the competition point, which is around 30 days after the establishment of bumblebee colonies [34]. At which time, workers begin to lay eggs, and WMs would be hatched after 26 days. Therefore, to ensure the purity of QMs, the QMs used in every experiment were enclosed from the switch point to 56 days after the establishment of the colony. Finally, five other colonies were exclusively reared to produce virgin queens for copulations.

### 2.2. Body Size, Sperm Quantity, and Sperm Viability of Males

From paired male producing colonies, 30 male bees from each of three queenright colonies and 30 male bees from each of the three corresponding queenless colonies (composed of 12 micro-colonies) were used for the measurement of body size, sperm quality, and mating ability. The other seven queenright colonies and 28 micro-colonies were prepared to produce sufficient QMs and WMs for competitive copulations.

Nine-day-old males, by which time they were sexually mature [34], were measured for body size, sperm quantity, and viability. In total, 90 WMs and 90 QMs from three paired queenless and queenright colonies were collected. The body size was first measured along with the weight by 1/10,000 electronic scale and the length of the left forewing by a digital caliper. Once measured, WMs and QMs were then dissected by scalpel under a stereomicroscope (Olympus SZX16, Tokyo, Japan) (Figure A1b). The accessory testes were obtained and quickly transferred to 400 μL buffer Kiev:M (1.67 mM glucose, 5.50 mM KCl, 2.50 mM NaHCO_3_, 8.26 mM trisodium citrate; pH 8.8). Sperm were then squeezed out gently by tweezers and mixed with buffer by gentle stirring. Half of the sperm solution was used for measuring the number of sperms using a cell counting chamber under a microscope (ZEISS scope. A1, Oberkochen, Germany). The cell counting chamber was composed of nine chambers, and the center chamber was the counting area with a 0.1 μL volume. The center chamber was divided into 25 small squares, and we counted the number of sperms in five squares as N so that the total number of sperms is N × 2 × 10^4^. Another half of the sperm solutions were used to determine sperm viability through the Live/Dead Sperm Viability Kit (Life Technologies Corporation, Carlsbad, America). The sperm solution was stained with SYBR−14 and propidium iodide, which marked live sperms as green color and dead sperms as red color, respectively. The live and dead sperms were counted in ten different fields of a fluorescence microscope (ZEISS scope. A1, Oberkochen, Germany) for each sample, and the sperm viability was figured out by the ratio of live sperms to total sperms.

### 2.3. Copulation and Mating Competition of the QMs and the WMs

Newly emerged QMs and WMs from three queenright and three queenless colonies composed of 12 micro-colonies, and queens from the five queen-producing colonies were collected daily and reared in plastic boxes separately. Nine-day-old males and ten-day-old queens were sampled for copulation experiments [34]. The copulation experiments were carried out in each colony: males from each colony and queens were placed in a proportion 2:1 (60 males and 30 queens) and were permitted to contact and mate freely in the mating cage (150 × 150 × 150 cm^3^). The mating cages were observed constantly, and mating pairs were transferred to separate boxes (Figure A1c). The number of WMs and QMs that copulated with queens successfully and the copulation duration of each mating pair was recorded. The mating ability of QMs and WMs were represented as the ratio of QM-mated and WM-mated queens to the total queens, respectively.

Moreover, we also set up the mating competition experiments to compare the competitiveness between QMs and WMs in copulating with queens. The experimental QMs, WMs were collected in other seven queenright and 28 corresponding queenless micro-colonies, and QMs from different colonies were mixed and reared for nine days, so were the WMs. Then QMs and WMs were marked on the mesonotum with different colors for easy identification. Two competitive copulation experiments were then undertaken, one where QM and WM were mixed in a 1:1 ratio, and another where they were mixed in a 9:1 ratio. The latter ratio is based on, but slightly higher than the previously reported ratio of WM in *B. terrestris* colonies (5% of males) [18]. Each competitive copulation group had three replicates. For each mating batch, 56 and 30 virgin queens were used in the 1:1 (100 QMs and 100 WMs) and 9:1 (108 QMs and 12 WMs) competitive copulation experiments, respectively. The layouts of every copulation experiment are shown in Table 1. All the copulation experiments were performed in a mating cage (150 × 150 × 200 cm^3^). In each copulation, mixed males were put into the mating cage firstly, and all the queens were then put into the cage at the same time to ensure an equal chance of mating. The mating cages were also observed constantly, and the queens that mated with QMs and WMs were transferred to separate boxes. The numbers of WM- and QM- mated queens were counted to evaluate the competitiveness of WMs. At the end of each mating experiment, the mated queens were reared, unmated queens, and all the males were abandoned. All the mated queens were fed pollen paste and sugar syrup for seven days and then kept in a 3 ± 1 °C refrigerator for three months diapause from December 2020 to March 2021.

### 2.4. Mated Queen Rearing and Colony Development

After the diapause, 122 QM-mated queens and 99 WM-mated queens were resuscitated at room temperature. In total, 83 QM-mated queens and 86 WM-mated queens survived and were reared in initial small boxes to lay eggs in controlled conditions of 28 °C and 60% RH (Figure A1d). The survival rate of mated queens was assessed as the ratio of survived mated queens to total queens. The egg-laying rate was assessed as the ratio of egg-laying queens to total reared queens within 20 days. Queens who had laid eggs were fed continually. The colony establishment time was recorded as the time when the first batch of workers emerged. Then the colonies were transferred to larger hives after the first batch of workers emerged. After 30 days, the number of offspring workers in each colony founded by WM- and QM- mated queens were counted. At the end of the colony life cycle, the number of offspring queens was also recorded.

### 2.5. Statistical Analysis

All data were analyzed using IBM SPSS20 (Chicago, IL, USA). Normality and homogeneity variance of all data sets were checked using the Shapiro–Wilk and Levene test. First, we investigated whether the body size differed significantly between the QMs and the WMs. Differences in weight and forewing length from two groups were compared and analyzed with *t*-tests, and the degree of dispersion of data of weight and forewing length was compared using the respective coefficients of variation. Spearman rank correlation test was used to describe the correlation between weight and forewing length of all the males because the data were not normally distributed. Moreover, *t*-tests were used to compare the sperm quantity and sperm quantity per mg body weight between QM and WM. As the data of sperm viability was not normally distributed, a nonparametric Mann–Whitney U-test was used to compare the sperm viability of QM and WM. Spearman rank correlation test and Pearson correlation test were used to analyze the correlations between sperm variability, sperm quantity, and weight/forewing length of QMs and WMs, respectively.

To evaluate the mating ability of WM, the Pearson chi-square test was used to compare differences between QM and WM based on the data of mated queens. A *t*-test was used to compare the differences in mating duration after applying a square root transformation of the data to comply with the assumption of normality prior to analysis. The preferences of queens for males in mating competitions with QM and WM in the proportion of 1:1 and 9:1 were analyzed with a one-sample chi-square test.

Finally, the survival rate of mated queens after diapause and laying rate after 20 days of rearing was evaluated using the Pearson chi-square test, the differences of populations between QM-mated and WM-mated queen colonies were analyzed using *t*-test. The coefficient of variation was introduced to compare the degree of dispersion of data of the number of workers and queens from QM-mated and WM-mated queen colonies.

## 3. Results

### 3.1. The WMs Possess a Smaller Body Size Than QMs

In total, weight and forewing length of 90 QMs and 88 WMs were measured and analyzed. The weight of QMs (mean ± SD: 371.07 ± 63.84 mg) were 37.04% heavier than that of WMs (mean ± SD: 256.99 ± 68.08 mg) (Figure 1A, *t* = 11.855, *p* < 0.000). Furthermore, the length of the forewing of QMs (mean ± SD: 15.91 ± 0.60 mm) was also significantly longer than that of WMs (mean ± SD: 13.57 ± 1.06 mm) (Figure 1B, *t* = 18.155, *p* < 0.000). Therefore, the WMs’ lighter weight and shorter forewing indicate a smaller body size compared to the QMs. Across all tested males (*n* = 178), weight showed a significant positive correlation with the forewing length (Figure 1C, ρ = 0.806, *p* < 0.000), while the coefficient of variation of male’s weight and forewing length was 28.2% and 9.8%, respectively. As such, forewing length can be an accurate proxy for evaluating the body size of bumblebee males.

### 3.2. The WMs Share Equal Sperm Viability and Slightly Less Sperm Quantity with the QMs

After measurement of weight and forewing length, 84 QMs and 88 WMs were chosen to evaluate the sperm quantity and sperm viability. Sperm quantity ranged from 1.32 to 0.08 × 10^6^ in QMs and 1.14 to 1.92 × 10^6^ in WMs, but WM showed a significantly lower mean sperm count (mean ± SD: 1.14 ± 0.29 × 10^6^) than that of QM (mean ± SD: 1.32 ± 0.26 × 10^6^) (Figure 2A, *t* = 4.359, *p* < 0.000). However, the mean sperm counts per mg body mass of WM (4.69 × 10^3^/mg) was significantly higher than that of QM (3.66 × 10^3^/mg) (Figure 2B, *t* = 5.984, *p* < 0.000). More remarkably, WMs shared equal sperm viability with QM, which was more than 90% on average (Figure 2C, Mann–Whitney U-test: U = 4421.500, *p* = 0.026). There is no correlation between sperm viability and body size based on the tested males (Figure 2D). Instead of sperm viability, sperm quantity shows a positive correlation with body size according to all tested males (Figure 2E).

### 3.3. The WMs Were Successful in Competitive Copulation with the QMs

In the current study, the mating rate and copulation duration were used to assess the mating ability of the QMs and WMs. Based on the queen counts, the mean success rate of copulation contributed by QMs was as high as 98%, but WMs possessed a lower mean success rate of copulation of about 87.1% compared to QMs (Figure 3A). Furthermore, and the mean copulation duration of WMs (mean ± SD: 34.55 ± 10.56 min) was shorter than that of QMs (mean ± SD: 41.74 ± 12.71 min) (Figure 3B). Therefore, there is a significant difference between QMs and WMs in mating rate (Pearson chi-square test: χ^2^ = 9.272, df = 1, *p* = 0.002) and copulation duration (*t* = 3.945, *p* < 0.000). However, low success copulation rate and short copulation duration show no effects on the mating success of WMs when compared with QMs. Thus, WMs were able to act as a successful competitor in the copulation with queens. For unmated QMs and WMs mixed in the 1:1 group, the number of mated QMs and WMs was 69 and 74. For unmated QM and WM mixed in the 9:1 group, mated QMs and WMs were counted as 69 and 13. Therefore, no matter the ratio of unmated QMs and WMs were 1:1 or 9:1, the mean proportion of mated WMs was 51.82% and 16.07%, respectively, which were higher than expected value in theory (50% and 10%), though the difference is not statistically significant (one-sample chi-square test: χ^2^ = 0.175, *p* = 0.676, χ^2^ = 3.122, *p* = 0.077) (Figure 3C).

### 3.4. The WM-Mated Queens Possess Equivalent Ability of Colony Foundation to the QM-Mated Queens

Mated daughter queens need to go through the diapause phase before beginning the new life cycle. A total of 122 QM-mated and 99 WM-mated queens were stored for diapause, of which 83 QM-mated and 86 WM-mated queens survived to the end of the diapause phase. So, under diapause conditions, the survival rate of WM-mated queens (86.87%) was significantly higher than that of QM-mated queens (69.84%) (Figure 4A, Pearson chi-square test: χ^2^ = 9.170, *p* = 0.002). After rearing for about 20 days in the initial small box, 62 QM- and 58 WM-mated queens had laid eggs, respectively (Figure 4B). Therefore, WM-mated queens share nearly the same egg-laying rate and possess a similar ability to lay eggs compared to the QM-mated queens (χ^2^ = 0.184, *p* = 0.668).

Subsequent rearing showed that 59 QM-mated queens and 45 WM-mated queens had established colonies, a total of 46.83% of QM- and 46.46% of WM-mated queens, respectively. The period of colony foundation, from the beginning of rearing, was between 35 and 41 days for both QM and WM (Figure 5A). The offspring worker populations were counted before the competition points that happen between queen and daughter workers. There was no statistically significant difference in worker numbers between colonies founded by QM-mated queens (mean ± SD: 95.06 ± 21.38) and colonies founded by WM-mated queens (mean ± SD: 88.97 ± 19.79) (Figure 5B, *t* = 0.511, *p* = 0.234). At the end of the life cycle, the number of offspring queens counted varied widely in both QM- (49 to 237) and WM-mated queen colonies (10 to 192) (Figure 5C). The coefficient of variation of the number of offspring queens in QM-mated queen colonies was 31.0%, and in WM-mated queen colonies was 36.3%. The mean numbers of offspring queens were 142 in QM-mated queen colonies and 130 in WM-mated queen colonies, which exhibited no significant differences between QM- and WM-mated queen colonies (Figure 5C, *t* = 0.589, *p* = 0.364).

## 4. Discussion

Around 69 out of 90 species of social Hymenoptera exhibit worker reproduction behaviors, which, although averse to the interests of the queen [56], is beneficial to workers who are more related to their sons than to their brothers [13,14]. Understanding the impacts of this is important to understanding the species reproduction as workers can produce a notable proportion of males from queenright colonies [33]. In this study, we analyzed reproductive characteristics of males from reproductive daughter workers, including the body size, sperm quality, copulation with queens, and population of the offspring swarm, to evaluate the fecundity of these males in comparison with males from the mother queen.

Our study confirmed the existence of somatotype dimorphism in *B. terrestris*. Worker-born males (WMs) were significantly lighter in weight and had smaller wings than that of the queen-born males (QMs) (Figure 1). This is consistent with studies of honeybees that also demonstrate lighter weight and smaller forewing of the WMs [47]. Furthermore, also consistent with honeybees, the variation range in forewing length between different individuals is much smaller than that of weight between WMs and QMs, so forewing length would be a more accurate indicator for body size to distinct WMs and QMs in bumblebees [46].

In honeybees, smaller males are at a disadvantage in mating success compared to the large males [50] as few have strong enough flying capabilities to successfully mate with queens in the drone congregation area (DCA) where thousands of males gather [57]. Even if smaller males are a higher proportion of available honeybee males, they still have significantly lower mating success and a smaller share of paternity than larger males [51]. A previous study has shown that *B. terrestris* males with different body sizes may differ in mating success, and heavier males mate more swiftly and faster than lighter ones [55]. However, in this study, the percentage of mated queens in copulation with WMs or QMs were both higher than reported in previous studies, which was around 75% in *B. terrestris* (Figure 3) [54,58]. In contrast to honeybees, no matter the ratio of WMs and QMs (1:1 or 1:9), *B. terrestris* WMs exhibited a little higher mate success ratio than theoretical expectations (Figure 3). Moreover, the copulation duration of WMs (36 min on average) was a little shorter than QMs (44 min on average), most sperms transferred into the queen genital tract within the first 2 min, and the mating plug was transferred within 10–30 min. Thus, a relatively shorter copulation duration of WMs was still adequate to meet the mating needs and had no influence on the colony development (Figure 4 and Figure 5) [44,59]. Therefore, we conclude that the WMs, despite possessing smaller body sizes, are successful in the male-male competition in copulation with queens in *B. terrestris*.

Beyond mating success, sperm quality is a more important indicator of the reproductive ability of both honeybee and bumblebee males [39,43]. Our results showed that WMs had lower sperm counts than QMs, but the sperm counts were still equal with the magnitude of sperm counts in western honeybee *Apis mellifera* (Figure 2) [48]. However, the population produced by the queen in honeybees is several hundred times larger than bumblebees whose colonies are only a few hundred at most [60,61,62]. Therefore, although the WMs possessed lower sperm counts, they were sufficient to produce enough diploid offspring workers or queens and haploid males in an annual life cycle of mother queens (Figure 4 and Figure 5). Similar to results reported in honeybees, larger males had 8.2% more spermatozoa than small males, and the difference in the number of spermatozoa is not expected to affect the paternity share if excluding the difference of semen concentration between large and small males [48,52]. Further, we found that in *B. terriestris*, WMs produced 20% more sperm count per mg of body mass than QM, which further parallels observations in honeybees [63]. Therefore, the rearing investment per sperm is lower in WM than in QM, which could be a strategy for reproductive workers to minimize the investigation and maximize fitness.

The sperm competition occurs in polyandrous eusocial Hymenoptera such as honeybees, ants, and wasps [64]. Although no significant difference was found between large and small males in sperm viability, small honeybee males still remained a little behind large males in sperm competition [48,52]. Bumblebee species can be divided into polyandrous species, e.g., *B. hypnorum* and monogamous species such as *B. terriestris* [65], where there is no sperm competition in any single mated queen. Apart from successful copulation and sufficient sperm counts, WMs also share equal sperm viability with QMs, so WM-mated queens are able to compete with QM-mated queens in founding colonies and producing populations of offspring (Figure 4). WM-mated queens had a similar egg-laying rate, colony-building rate and produced similar numbers of workers and queens to QM-mated queens. In conclusion, WMs from dominant workers possess the inherent reproductive ability to compete with the QMs for mate success, colony establishment, and offspring populations in spite of the smaller body size and less sperm counts.

As WMs could threaten the heredity of the queen’s sons, the queen inhibits the production of WMs through secretion of pheromonal signaling, attacking behavior, worker policing, and so on [23,24,26,27,28]. Even so, approximately 5% of males are produced by reproductive workers in the queenright *B. terrestris* colony [18,19]. However, other bumblebee species may experience much greater worker and queen competition for males. For instance, over 20% of males are produced by reproductive workers in *B. hypnorum* queenright colony [66]. Most surprisingly, more than 50% of males were produced by workers in some species, such as the neotropical bumblebee *B. wilmattae* [67]. The phenomenon might be because social bumblebees are primitively eusocial species where caste determination is largely post-imaginal. Adults have flexible roles and can more readily switch from worker to queen roles [68]. Bumblebees are monandrous and facultative low polyandry species that have reproductive totipotent workers [69]. So, the males from workers might help the populations to minimize the negative effects of inbreeding and contribute to their overall reproductive success, as reported in eastern honeybee *Apis cerana* [70]. Moreover, queen mating frequencies and proportions of WMs might vary between different bumblebee subgenera [18,57,66,67,71,72]. Further studies are needed to explore the effect of WMs on the evolution of social insects, especially bumblebees. In addition, it is very difficult to recognize the WMs from the QMs in a given *B. terrestris* queenright colony. We obtained WMs from the queenless worker micro-colonies [53], which had been successfully used for investigating a range of endpoints, including behavior, gut microbiome, nutrition, development, pathogens, chemical biology, and pesticides/xenobiotics [73]. Otherwise, only dominant workers (usually old large ones) produce males in queenright *B. terrestris* colony [74,75], and usually, the largest one of workers was stimulated to establish dominance and begin laying eggs after separating from the queenright colony [76]. Our work also demonstrates that the queenless micro-colony is a suitable model with which to study bumblebee males.

## Figures and Tables

**Figure 1 insects-12-01008-f001:**
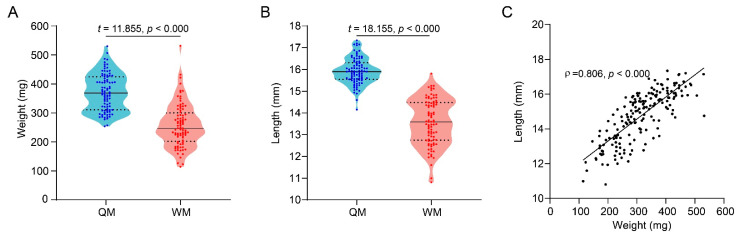
Comparison of body weight and forewing length of the QMs and the WMs. (**A**) Body weight of the QMs and the WMs (*n* = 90 and 88, respectively); (**B**) forewing length of the QMs and the WMs (*n* = 90 and 88, respectively); (**C**) correlation between weight and forewing length of all the experimental the QMs and the WMs (*n* = 178). QMs and WMs represent the males produced by queens in queenright colonies and by workers from queenless micro-colonies, respectively. A *t*-test was used to analyze whether significant differences existed in weight and forewing length between the QMs and the WMs. Spearman rank correlation test was used to analyze the correlation between weight and forewing length of male bees. The scatter plots represented the data values.

**Figure 2 insects-12-01008-f002:**
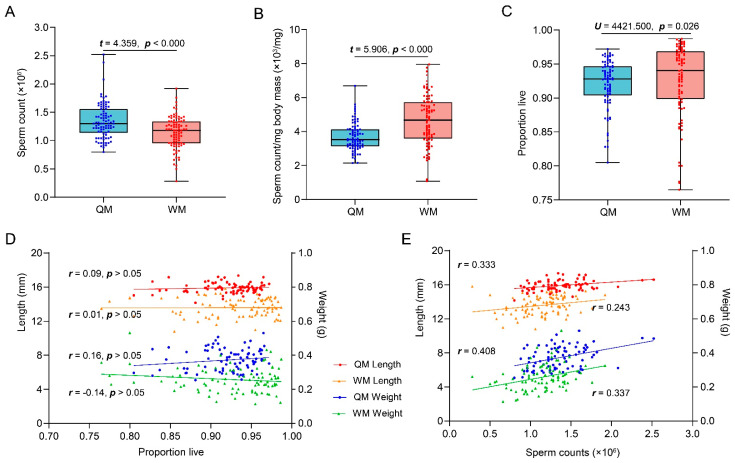
Comparison of sperm quantity, sperm quantity per mg body mass, sperm viability of the QMs and the WMs, and the correlation analysis between body size and sperm quality. (**A**) Sperm counts of the QMs and the WMs (*n* = 84, 88); (**B**) sperm counts per mg body mass of the QMs and the WMs (*n* = 84, 88); (**C**) the sperm viability of the QMs and the WMs (*n* = 84, 88); (**D**) the correlation between sperm viability and body size of the QMs and the WMs; (**E**) the correlation between sperm counts and body size of the QMs and the WMs. A *t*-test was used to analyze whether significant differences existed in sperm counts and sperm counts per mg body size between QM and the WM. The Mann–Whitney U-test was used to analyze whether significant differences existed in sperm viability between the QM and the WM. Spearman rank correlation test was used to analyze the correlation between sperm viability and body size, Pearson correlation test was used to analyze the correlation between sperm counts and body size. The scatter bots represent the data values, and the lower edge lines of the box represent the upper quartile and the lower quartile in (**A**–**C**). The dots represent the data values in (**D**,**E**).

**Figure 3 insects-12-01008-f003:**
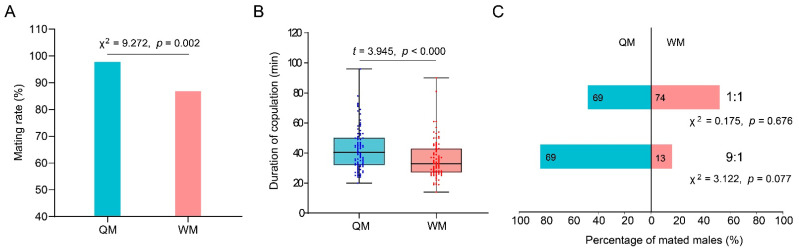
Comparison of mating ability and competitive copulation between the QMs and the WMs. (**A**) The mating rate of the QMs and the WMs based on the proportion of mated queens; (**B**) the duration of copulation contributed by QMs and WMs (*n* = 82, 79). The scatter plots represent the data values, and the upper and lower edge lines of the box represent the upper quartile and the lower quartile; (**C**) the percentage of mated QMs and WMs in competitive copulations in the 1:1 and 9:1 ratio experiments, respectively. The number in each bar represents the number of males that successfully mated with a queen. The Pearson chi-square test was used to analyze the differences in the mating rate of the QM and the WM. The t-test was used to analyze whether significant differences existed in mating duration. The one-sample chi-square test was used to analyze whether significant differences existed between the ratio of mated the QMs and the WMs and the theory proportion of the QMs and the WMs.

**Figure 4 insects-12-01008-f004:**
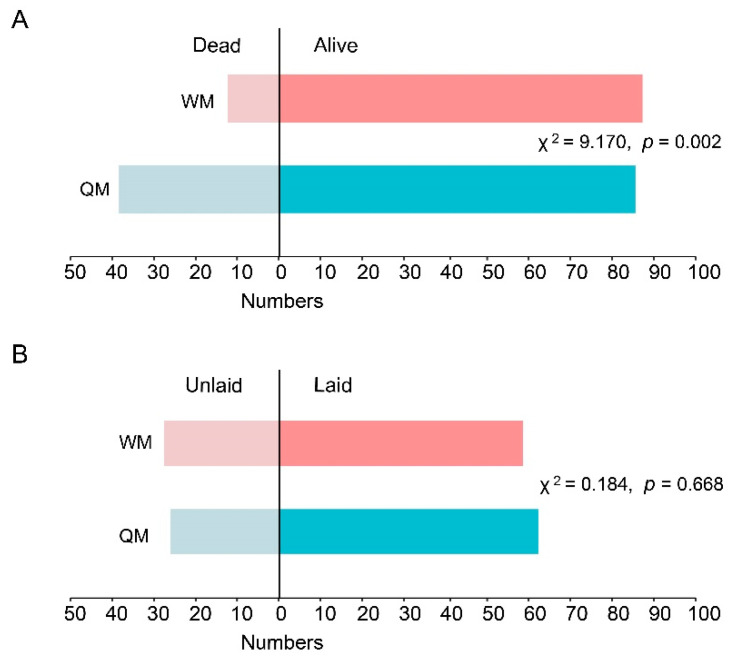
The survival rates and egg-laying rates of the QM- and the WM-mated queens. (**A**) The number of dead and alive QM- and WM-mated queens after diapause; (**B**) the number of laid and unlaid QM- and WM-mated queens after 20 days’ rearing. The Pearson chi-square test was used to analyze the differences in survival results of QM- and WM-mated queens after diapause and egg-laying results after 20 days’ rearing.

**Figure 5 insects-12-01008-f005:**
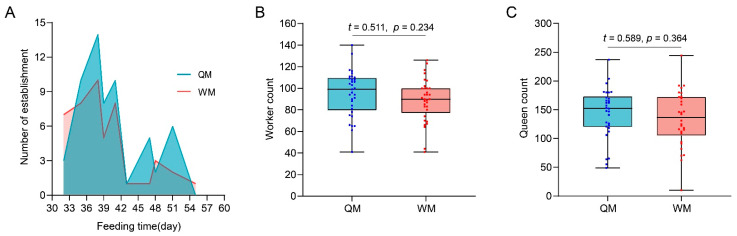
Colony foundation periods and offspring female populations of the QM- and the WM-mated queen colonies. (**A**) The colony established a process of QM- and WM-mated queen colonies. The two broken lines represent the feeding days for the establishment of colonies and the number of established colonies every day; (**B**) the numbers of offspring workers in QM- and WM-mated queen colonies after 30 days from the establishment of colonies (*n* = 29, 29); (**C**) the numbers of offspring queens in QM- and WM-mated queen colonies (*n* = 29, 29). The scatter plots represented the data values, and the upper and lower edge lines of the box blots represent the maximum and minimum values of data in (**B**,**C**). A *t*-test was used to analyze whether significant differences existed in the number of workers and daughter queens between QM- and WM-mated queen colonies.

**Table 1 insects-12-01008-t001:** The number of individuals in competitive copulation.

Proportion of QMs and WMs	Mating Batches	Number of Queens	Number of QMs	Number of WMs
1:1	1	56	100	100
	2	56	100	100
	3	56	100	100
9:1	1	30	108	12
	2	30	108	12
	3	30	108	12

The QMs and WMs are the abbreviations of queen-born males and worker-born males.

## Data Availability

The data presented in this study are available on request from the corresponding author.

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
