# Peer review of "Worker-Born Males Are Smaller but Have Similar Reproduction Ability to Queen-Born Males in Bumblebees"

_insects, 2021, doi:10.3390/insects12111008_

Round 1
Reviewer 1 Report
A solid paper that makes a valuable contribution to the field.
Although it may appear quite specific, it has broader relevance within the field of post-copulatory sexual selection and insect reproductive biology in general. The language needs fairly extensive review.
Author Response
Answer (A):
Dear Colleague, Many thanks for your positive comments. We have improved the language according to your good suggestions. Please see the revised version attached. We hope you will find the improvements to the manuscript satisfactory.

Reviewer 2 Report
The paper presented to me describes an original and neatly planned experiment aiming to compare reproductive success of worker-born and queen-born bumblebees. The experiment is well planned and allows to answer the research question of this study. I cannot pinpoint any weakness in the design of the experiment and fully agree with the data analysis and interpretation.
I only have a few really minor corrections and questions to the Authors:
#74 mellifera not millifera
#120-122 You mean here, that you have collected males that were laid between the 23-26 day after the colony started to ensure no WMs were mixed into the QM group
# 171-178 Can you describe more in detail the how the competitive copulation was done? You put together eg. 56 queens with 200 males form the two groups? Or one queen with ten males at once? What do you mean by biological replicate? One queen used 3x or something else?
Author Response
Answer (A):
Dear Colleague, Many thanks for your positive comments and good suggestions for the manuscript. We have improved the language according to your suggestions, and have addressed three points you recommended. Please see the revised version attached. We hope you will find the improvements to the manuscript satisfactory.
I only have a few really minor corrections and questions to the Authors:
#74 mellifera not millifera
A: Name has been corrected.
#120-122 You mean here, that you have collected males that were laid between the 23-26 day after the colony started to ensure no WMs were mixed into the QM group
A: QMs were produced from the switch point to the end of the colony. WM eggs were laid from the competition point (about 30 days after the establishment of colony) and it took 26 days for males to emerge. So, around 56 days after the establishment of colony, males in the colony were the mixture of QMs and WMs. Therefore, to ensure the purity of QMs, the QMs used in every experiments were enclosed from the switch point to 56 days after the establishment of bumblebee colonies.
We have improved the description on lines 129-141.
# 171-178 Can you describe more in detail the how the competitive copulation was done? You put together eg. 56 queens with 200 males form the two groups? Or one queen with ten males at once? What do you mean by biological replicate? One queen used 3x or something else?
A: There are two groups of competitive copulation with males mixed in proportion 1:1 or 9:1 and each group was repeated for three times. In each mating experiment, we put all mixed males (200 in 1:1 and 120 in 9:1) in the mating cage, and then put all queens together into the cage ( 56 in 1:1 and 30 in 9:1) in competitive copulation. All males and queens were used only once whether they had successfully mated or not. A total of 168 queens were used in the proportions 1:1 and 90 queens in 9:1.
We have improved the description on lines 187-212.
